# Transient auditory nerve demyelination as a new mechanism for hidden hearing loss

Guoqiang Wan[1,2] & Gabriel Corfas[1]

Hidden hearing loss (HHL) is a recently described auditory neuropathy believed to contribute to speech discrimination and intelligibility deficits in people with normal audiological tests. Animals and humans with HHL have normal auditory thresholds but defective cochlear neurotransmission, that is, reduced suprathreshold amplitude of the sound-evoked auditory nerve compound action potential. Currently, the only cellular mechanism known for HHL is loss of inner hair cell synapses (synaptopathy). Here we report that transient loss of cochlear Schwann cells results in permanent auditory deficits characteristic of HHL. This auditory neuropathy is not associated with synaptic loss, but rather with disruption of the first heminodes at the auditory nerve peripheral terminal. Thus, this study identifies a new mechanism for HHL, highlights the long-term consequences of transient Schwann cell loss on hearing and might provide insights into the causes of the auditory deficits reported in patients that recover from acute demyelinating diseases such as Guillain–Barré syndrome.

[1] Department of Otolaryngology–Head and Neck Surgery, Kresge Hearing Research Institute, University of Michigan, 1150 West Medical Center Drive, Ann Arbor, Michigan 48109, USA. [2] MOE Key Laboratory of Model Animals for Disease Study, Model Animal Research Center of Nanjing University, Nanjing, Jiangsu Province, China. Correspondence and requests for materials should be addressed to G.W. (email: wangq@nicemice.cn) or to G.C. (email: corfas@med.umich.edu).

In the mammalian cochlea, acoustic stimuli are detected by inner hair cells (IHCs), which transduce and transform them into synaptic signals to the primary auditory sensory neurons, the spiral ganglion neurons (SGNs). These neurons then relay the acoustic information to the central auditory circuits. Thus, proper function of both sensory cells and neurons is essential for normal hearing and auditory communication, and degeneration of either component causes sensorineural hearing loss. This type of hearing loss, which affects 320 million people worldwide[1], is characterized by permanent elevation of auditory thresholds, that is, the minimal sound pressure levels (SPLs) that evoke an auditory or neural response at specific sound frequencies. Until recently, it was believed that auditory processing is normal in subjects with normal thresholds, even after recovery from a temporary threshold shift (TTS) due to noise exposure, and that hearing deficits in those cases were due to central problems[2]. However, recent animal and human studies indicate that moderate noise exposures or ageing can result in a new form of peripheral hearing loss called 'hidden hearing loss (HHL)'[3–6]. HHL can be detected physiologically and is characterized by normal auditory thresholds but reduced suprathreshold amplitude of the sound-evoked SGN compound action potential (AP) (the first peak of the auditory brainstem response (ABR) waveform)[3,5–8], and by the alteration in the ratio between the peak of the ABR waveform generated by hair cells (summating potential (SP)) and the peak I of the ABR waveform (AP)[9]. The reduced neuronal activation seen in HHL has been proposed to result in degradation of the temporal coding of suprathreshold sounds and deficits in speech discrimination and intelligibility, particularly in a noisy environment[10,11]; however, the latter are not diagnostic of HHL, as they could be also caused by central processing deficits. HHL has been proposed to lead to hyperactivation of central auditory synapses, thus possibly contributing to tinnitus[3,12]. Until now, loss or dysfunction of IHC synapses has been the only proposed mechanism of HHL[13]. Here we report an additional cellular mechanism for HHL.

In the cochlea, sensory hair cells and neurons are in close association with several types of glial and glia-like cells[14]. Hair cells are surrounded by supporting cells, SGN axons are myelinated by Schwann cells and spiral ganglion cell bodies are wrapped by satellite cells (Fig. 1a). In prior studies we explored the roles of supporting cells in IHC synapse formation and regeneration[15] and in the preservation of hair cells[16]. As part of our ongoing efforts to define the roles the different populations of cochlear glia play in hearing and deafness[15–17], we investigated the consequences of Schwann cell ablation in the mature cochlea. We found that acute Schwann cell loss causes rapid auditory nerve demyelination, which is followed by robust Schwann cell regeneration and axonal remyelination. Surprisingly, this transient Schwann cell loss results in a permanent auditory impairment characteristic of HHL. However, this HHL differs from that seen after noise exposure or ageing in that it occurs without alterations in synaptic density but rather correlates with a specific and permanent disruptions of the first heminodes at the auditory nerve axon close to the IHCs. Furthermore, the two types of HHL occur independently (noise exposures that cause HHL do not disrupt heminodes) and they are additive. Together, these results uncover a new mechanism for the pathogenesis of HHL and a new consequence of myelin defects on the normal function of the nervous system.

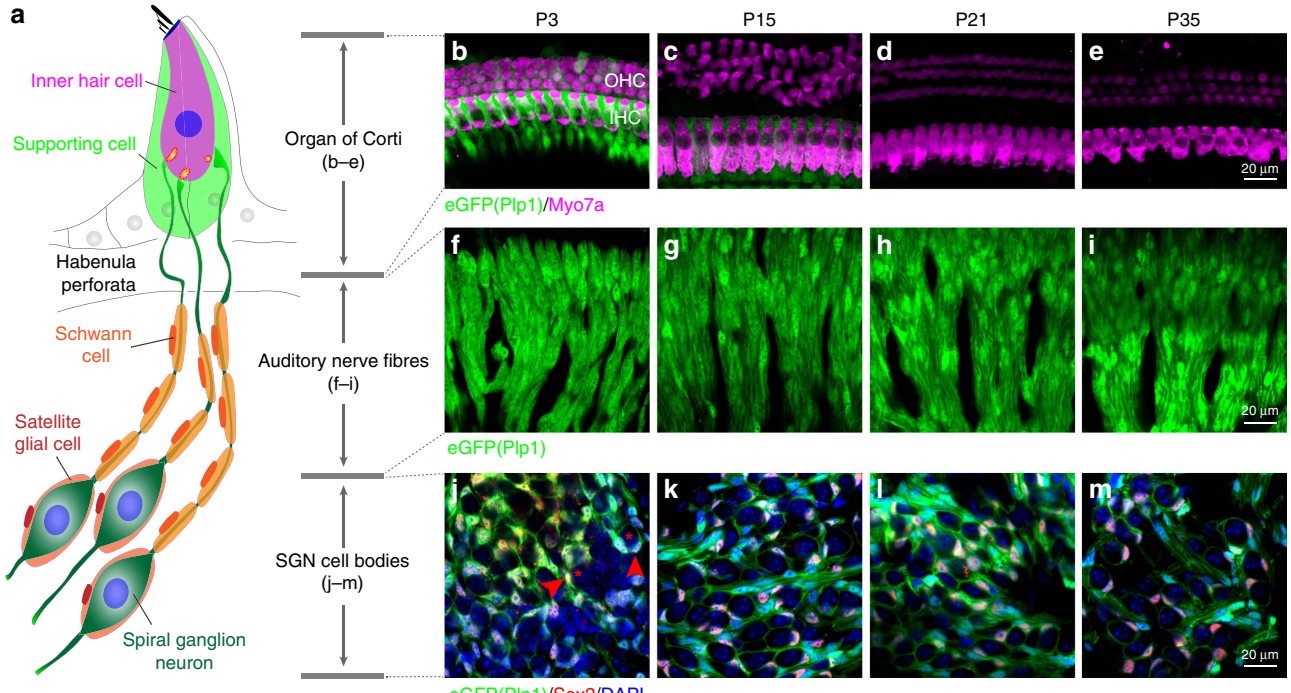

**Figure 1 | *Plp1* is expressed only by Schwann and satellite cells in the mature cochlea. (a)** Schematic illustration of the IHC, SGNs and their associated glial/glial-like cells. (**b–e**) Images of the organ of Corti of *Plp1/eGFP* mice at different postnatal ages (P3–P35) show that the *Plp1* promoter is active in supporting cells associated with IHCs until P15, but by P21 is silent. Three rows of outer hair cells (OHCs) and one row of IHCs were labelled by Myo7a. (**f–i**) Images of cochlear whole mounts of *Plp1/eGFP* mice show that the *Plp1* promoter is active in auditory nerve Schwann cells in the OSL at all ages examined. (**j–m**) Images of sections through the spiral ganglion of *Plp1/eGFP* mice show that the *Plp1* promoter is active in satellite cells surrounding SGNs at all ages examined. Sections were co-labelled with a marker for glial cells (Sox2) and nuclei (4,6-diamidino-2-phenylindole, DAPI). Asterisks and arrow heads indicate SGN cell bodies and associated satellite cells, respectively.

## Results

**Ablation and regeneration of Schwann cells in mature cochlea.**
To test the consequence of Schwann and satellite cell ablation on the cochlea, we used conditional expression of diphtheria toxin fragment A (DTA) in proteolipid protein 1 (*Plp1*)-expressing cells at P21. This strategy was based on the pattern of *Plp1* promoter activity in the postnatal cochlea. As shown in Fig. 1, in the neonatal mouse cochlea (P1–P15) *Plp1* is expressed by Schwann cells, satellite cells, as well as inner phalangeal and inner border cells (IPhC/IBCs), the supporting cells surrounding IHCs[18]. However, by P21 *Plp1* promoter activity in IPhC/IBCs is undetectable (Fig. 1a–e), while is maintained in Schwann cells along the auditory nerve fibres (ANFs) located at the osseous spiral lamina (OSL) (Fig. 1a,f–i) and satellite cells surrounding the cell bodies of SGNs in Rosenthal's canal (Fig. 1a,j–m).

As anticipated from prior studies of mice with DTA-induced ablation of *Plp1*-expressing cells[19], tamoxifen intraperitoneal (i.p.) injection to mice carrying *Plp1/CreER$^T$* transgene and *Rosa26DTA* alleles [*DTA(+/−):Plp1/Cre(+/−)*] from P21 to 23 resulted in uncoordinated gait and hind-limb paralysis between 1 and 2 weeks post DTA expression followed by complete recovery of normal gait by 3–4 weeks post DTA expression. This is consistent with the robust regenerative capacity of *Plp1*-expressing glia cells[19]. Interestingly, although DTA induction at P35 results in significant mortality 3 weeks later (ref. 19 and our unpublished data), we found that ablation of

*Plp1*-expressing cells at P21 does not result in lethality. These observations suggest that the health effects of loss of *Plp1*+ cells increase with age.

Based on the findings described in Fig. 1, we anticipated that tamoxifen treatment of *DTA(+/−)/PlpCre1(+/−)* mice from P21 to P23 would ablate Schwann and satellite cells without affecting the supporting cells in the organ of Corti. Indeed, immunostaining of cochlear whole mounts with the Schwann cell marker Sox10 or myelin basic protein (MBP) showed that *DTA* transgene activation at P21 results in robust ablation of Schwann cells along the ANFs within 1 week after tamoxifen injection (Supplementary Fig. 1a–d). In contrast, IPhC/IBCs, which are labelled by Sox2 or GLAST, are spared from DTA-mediated ablation at this age (Supplementary Fig. 1e,f). With this model in hand, we analysed the effects of the Schwann cell loss on cochlear structure and function at different intervals, from 1 week to 1 year after induction of Schwann cell ablation.

Studies of other peripheral nerves have demonstrated that demyelination and Schwann cell loss is usually followed by extensive proliferation of undifferentiated Schwann cells, which repopulate and remyelinate the nerves[20,21]. Accordingly, analysis of 5-ethynyl-2′-deoxyuridine (EdU) incorporation showed that 1 week after *DTA* transgene activation, the auditory nerves of mice with Schwann cell ablation display a dramatic increase in the number of EdU+ cells (Fig. 2c,d) compared with controls (Fig. 2a,b). Furthermore, lineage tracing with an inducible

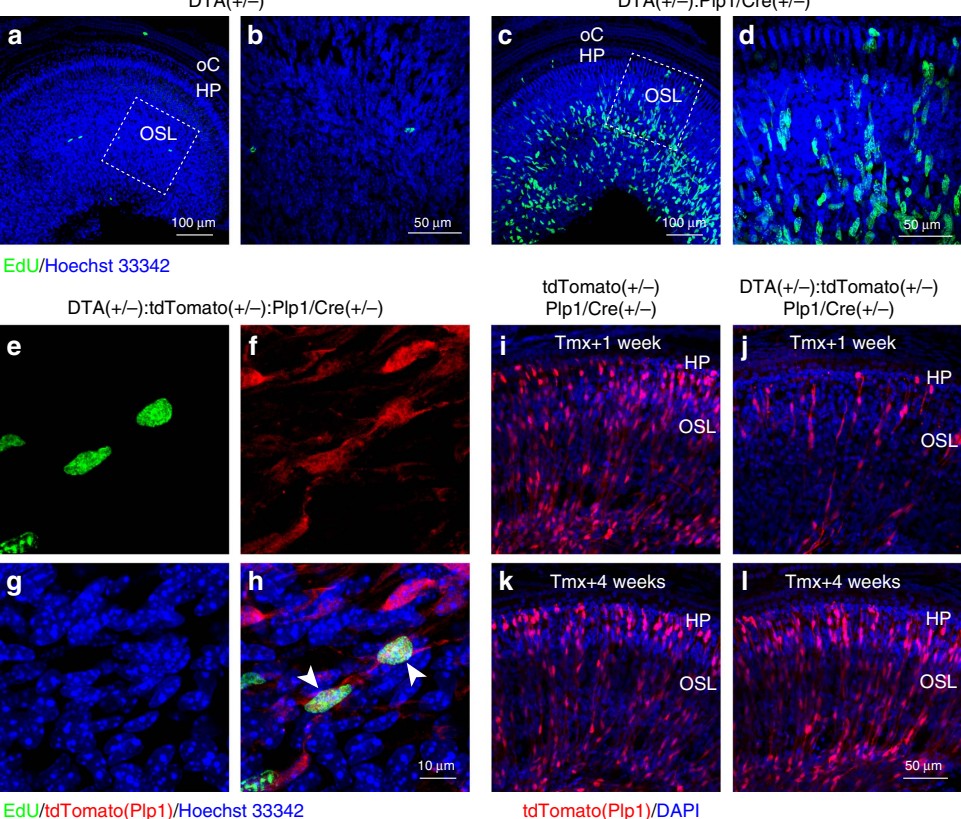

**Figure 2 | Schwann cells in the auditory nerve regenerate within weeks after DTA induction in *Plp1*-expressing cells.** (**a–d**) Tamoxifen induction of *DTA* expression at P21 promotes EdU incorporation in *DTA(+/−):Plp1/Cre(+/−)* cochlea (**c,d**), but not the control cochlea (**a,b**). Mice were injected with tamoxifen at P21–23, with EdU at P29 and tissues harvested at P30. (**b,d**) High-magnification views of the dashed boxes in **a** and **c**, respectively. HP, habenula perforata; oC, organ of Corti; OSL, osseous spiral lamina. (**e–h**) Lineage tracing of EdU+ cells shows that they are Schwann cells that escaped DTA-mediated death. (**h**) Merged image of EdU labelling (**e**, green), *Plp1*-driven tdTomato (**f**, red) and nuclei (**g**, blue). Arrow heads in **h** indicate surviving *Plp1*-expressing cells incorporated EdU. (**i–l**) tdTomato-positive cells repopulate in 4 weeks after tamoxifen induction. Control (**i,k**) and *DTA(+/−):tdTomato(+/−):Plp1/Cre(+/−)* mice (**j,l**) were injected with tamoxifen from P21 to 23 and cochlear tissues analysed either 1 week (**i,j**, Tmx+1 week) or 4 weeks (**k,l**, Tmx+4 weeks) later.

tdTomato reporter in *DTA(+/−):tdTomato(+/−):Plp1/Cre(+/−)* mice revealed that these EdU-positive cells are generated by *Plp1*-expressing cells that most probably escaped DTA-mediated ablation (Fig. 2e–h), that is, Schwann cells in which recombination occurred in the *Rosa26tdTomato* allele but not in the *Rosa26DTA* allele. Further analysis showed that the surviving tdTomato+ cells 1 week post induction (Fig. 2i,j) repopulate the OSL to levels similar to that of controls by 1 month post induction (Fig. 2k,l).

Similar robust regeneration was observed for satellite cells (Supplementary Fig. 2). Although satellite cells are clearly lost 1 week post induction (Supplementary Fig. 2a,b), they repopulate within 2 weeks post induction in *DTA(+/−):Plp1/eGFP(+/−):Plp1/Cre(+/−)* cochlea (Supplementary Fig. 2c,d). Plastic sections of Rosenthal's canal 16 weeks post induction show that both the SGN cell body density (Supplementary Fig. 2e–g) and the percentage of SGN cell bodies wrapped by satellite cells (Supplementary Fig. 2e,f,h) are similar in *DTA(+/−):Plp1/Cre(+/−)* and control cochlea. Together, these results

indicate that cochlear Schwann and satellite cells spontaneously and robustly repopulate after DTA-mediated ablation in the mature cochlea, and that Schwann cell regeneration is at least partly mediated by proliferation of the surviving Schwann cells.

**Schwann cell regeneration and remyelination of SGN axons.** In models of axon injury or demyelinating neuropathy, damaged Schwann cells regenerate and re-myelinate axons[20,22,23]. We therefore examined the consequence of DTA-induced Schwann cell ablation on ANF myelin. *DTA* expression resulted in acute demyelination of the ANFs within 1 week post induction (Fig. 3a,b), consistent with loss of Schwann cell markers at this time point (Supplementary Fig. 1). At 4 weeks post induction, when Schwann cells have repopulated (Fig. 2i,j), we observed significant yet incomplete remyelination (Fig. 3a,b). By 16 weeks post induction, myelinated axon density completely recovered in *DTA(+/−):Plp1/Cre(+/−)* cochleas (Fig. 3a,b). Moreover, at 16 weeks post induction, axons from *DTA(+/−):Plp1/Cre(+/−)* cochleas had

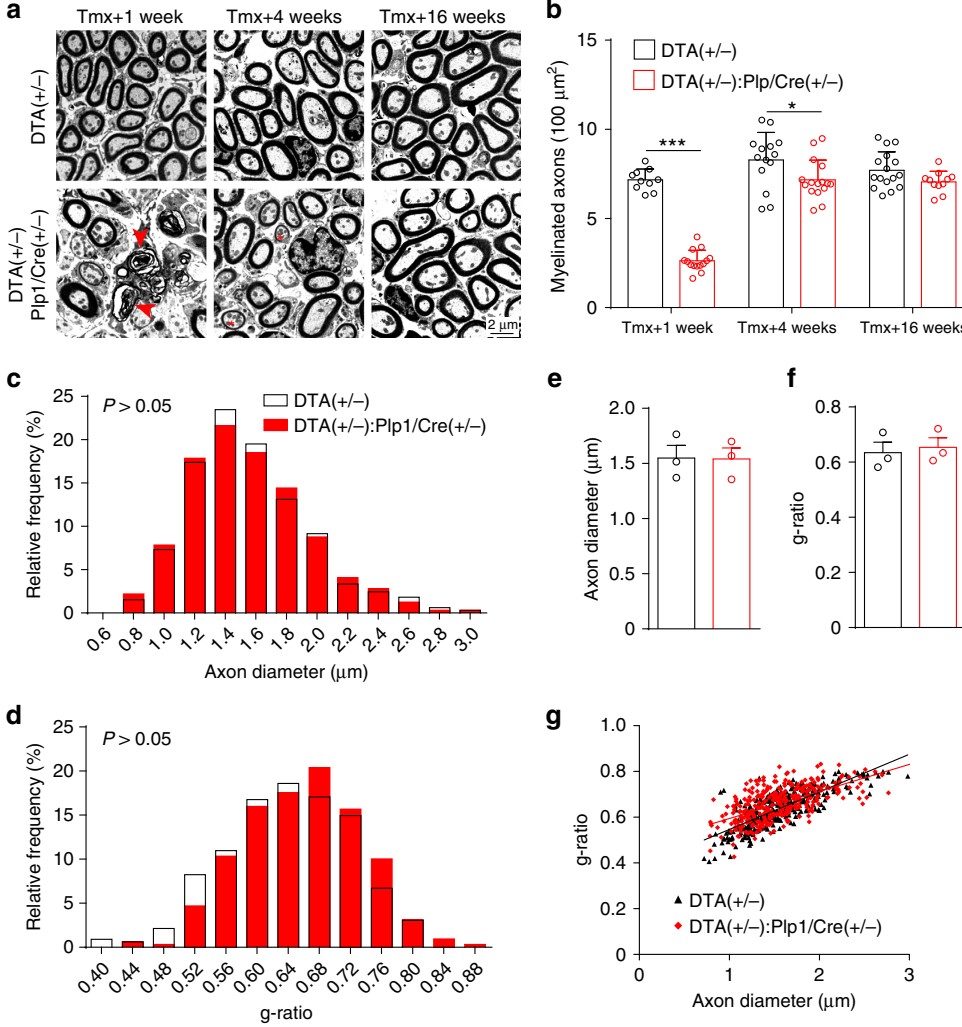

**Figure 3 | Transient Schwann cell ablation causes acute demyelination followed by remyelination of the ANFs. (a,b)** Representative electron micrographs of sections through the OSL show ANF in *DTA(+/−):Plp1/Cre(+/−)* cochlea are demyelinated at 1 week after tamoxifen injection, followed by partial remyelination 4 weeks later and complete remyelination at 16 weeks. Arrowheads show pathological myelin sheaths and asterisks show partially re-myelinated axons. **(b)** Quantitative analysis of myelinated axon density in the OSL at different times after tamoxifen treatment; $n = 10$–16 from 3 individual animals in each group. *$P < 0.05$ and ***$P < 0.001$ by two-way analysis of variance (ANOVA) followed by Bonferroni's post-test. **(c–g)** ANFs of *DTA(+/−):Plp1/Cre(+/−)* cochlea have normal axon caliber and myelin thickness 16 weeks after tamoxifen induction. Axonal diameter **(c,e)** and g-ratio **(d,f,g)** are normal in *DTA(+/−):Plp1/Cre(+/−)* cochleae 16 weeks after tamoxifen treatment. g-ratio was calculated as axon diameter / (axon + myelin sheath diameter). For **e–g**, $n = 3$ animals of each group with $> 300$ axons per group. Data are expressed as mean ± s.d.

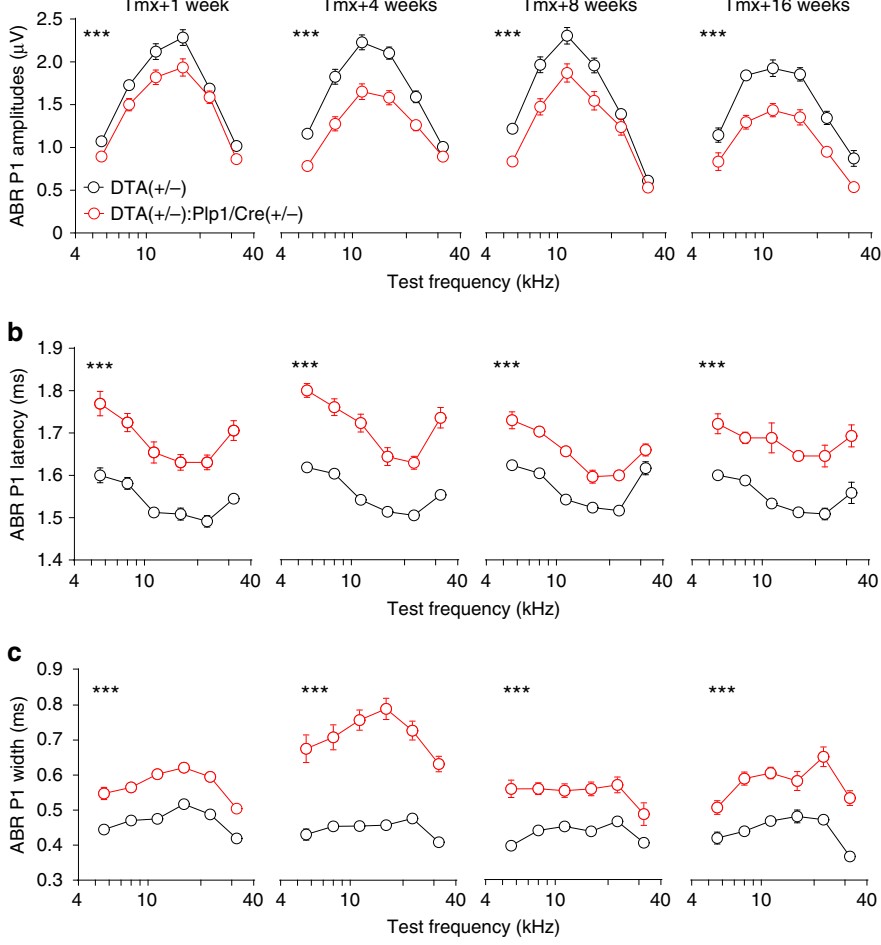

**Figure 4 | Transient Schwann cell ablation results in persistent impairment of auditory physiology.** Mice were injected with tamoxifen from P21 to 23 and ABR tests were performed 1, 4, 8 and 16 weeks later. At all time points examined, ABR P1 supra-threshold amplitudes (**a**) are reduced, ABR P1 latencies (**b**) are increased and ABR P1 widths (**c**) are increased in $DTA(+/-):Plp1/Cre(+/-)$ mice. ABR P1 amplitudes, latencies and widths were analysed at 70 dB SPL. Tmx + 1 week, $n = 17$–21; Tmx + 4 weeks, $n = 16$–20; Tmx + 8 weeks, $n = 10$–14; Tmx + 16 weeks, $n = 7$–8 of each animal group. ***$P < 0.001$ by two-way ANOVA. Data are expressed as mean ± s.e.m.

normal axon diameters (Fig. 3c,e,g) and myelin thickness (Fig. 3d,f,g). These results indicate that acute Schwann cell ablation results in rapid cell regeneration followed by remarkable remyelination, leading to fibres with normal axon caliber and myelin thickness.

**Transient Schwann cell ablation causes permanent HHL.** Conduction block is the fundamental finding in clinically important acquired demyelinating neuropathies such as acute inflammatory demyelinating polyneuropathy and chronic inflammatory demyelinating polyneuropathy[24]. Therefore, given the dramatic demyelination we observed in ANFs on activation of the *DTA* transgene, we anticipated that mice would display significant hearing abnormalities soon after induction, and that these defects would resolve within 4 months as the nerve remyelinated. However, thresholds for the ABR and distortion product otoacoustic emission (DPOAE), which reflect the function of IHCs and outer hair cells, respectively, were not affected by the lack of myelin in ANFs (Supplementary Fig. 3a,b), indicating that these mice were 'audiologically normal.' In contrast, the suprathreshold amplitudes of the ABR peak 1 (P1 amplitudes), which represent the summed activity of the SGNs, were significantly reduced (Fig. 4a and Supplementary Fig. 4). Furthermore, the suprathreshold latencies and widths of the ABR peak 1, which reflect in part the conduction velocity of the

auditory nerve, were both severely affected (Fig. 4b,c and Supplementary Fig. 5). Specifically, mice with DTA-induced Schwann cell ablation have increased P1 latencies (Fig. 4b and Supplementary Fig. 5) and P1 widths (Fig. 4c) at all cochlear frequencies examined (5.6–32 kHz). Surprisingly, the effects of Schwann cell ablation on P1 amplitudes, latencies and widths do not reverse even 16 weeks post induction, when Schwann cells and myelin have regenerated (Figs 2 and 3 and Supplementary Fig. 2). Further support to the conclusion that the transient demyelination results in HHL was provided by analysis of the SP/AP ratios (amplitudes of SP/amplitudes of AP) (Fig. 5a), which can be used as a differential diagnosis for HHL in both mice and humans[5,9]. Transient Schwann cell ablation did not affect SP at all time points (Fig. 5b), further demonstrating that Schwann cell loss did not alter hair cell function (Supplementary Figs 1 and 3). In contrast, peak 1 AP amplitudes in $DTA(+/-):Plp1/Cre(+/-)$ mice were reduced at all time points (Fig. 5c), suggestive of cochlear neural dysfunction. Furthermore, SP/AP ratios were significantly reduced 4–16 weeks after Schwann cell ablation (Fig. 5d). These findings indicate that transient Schwann cell ablation results in prolonged auditory neuropathy and HHL.

**Demyelination-induced HHL is not caused by synapse loss.** Loss of synapses between IHCs and SGNs is currently the only

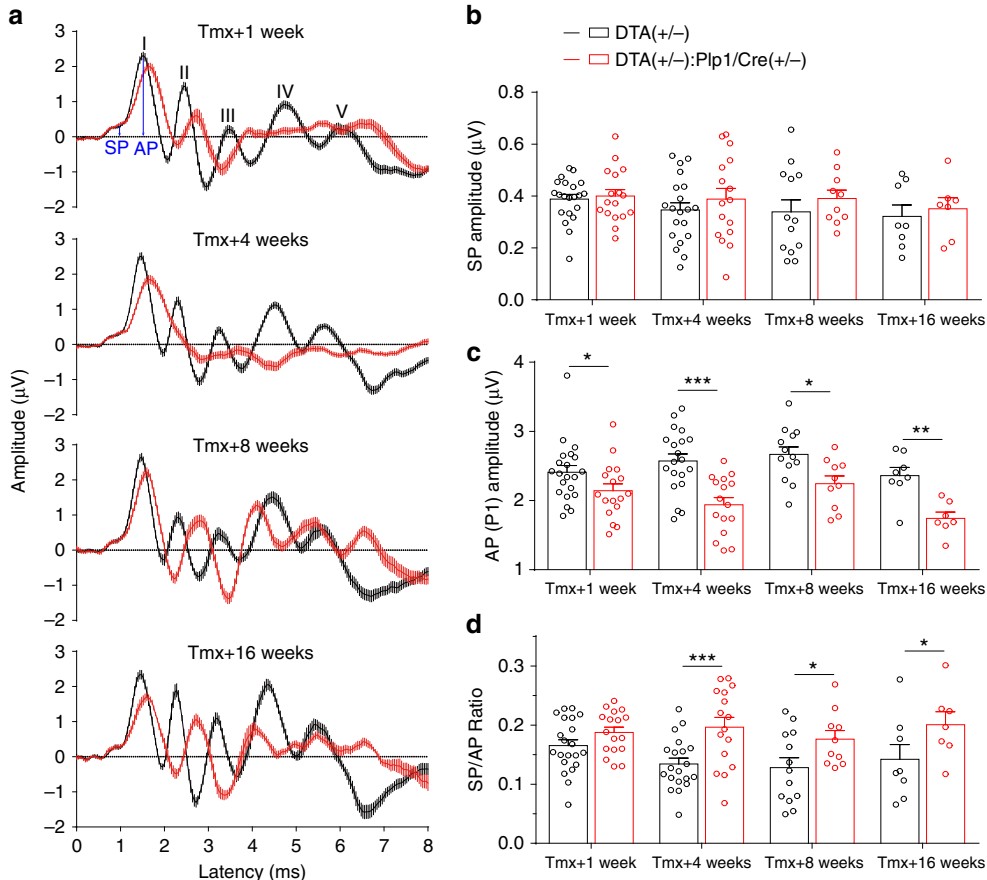

**Figure 5 | Auditory APs and SPs are differentially affected by transient Schwann cell ablation.** Mice were injected with tamoxifen from P21 to 23 and ABR tests were performed 1, 4, 8 and 16 weeks later. (**a**) Averaged ABR traces of control and *DTA(+/−):Plp1/Cre(+/−)* mice at 11.3 kHz, 70 dB SPL. Five ABR peaks are labelled with I–V. Arrows indicate SP and ABR P1 AP. (**b**) The SP of *DTA(+/−):Plp1/Cre(+/−)* mice are not affected at all time points examined. (**c**) ABR P1 AP were reduced in *DTA(+/−):Plp1/Cre(+/−)* mice at all time points examined. (**d**) Ratios of SP/AP are increased in *DTA(+/−):Plp1/Cre(+/−)* mice 4, 8 and 16 weeks after tamoxifen induction. Tmx + 1 week, $n = 17–21$; Tmx + 4 weeks, $n = 16–20$; Tmx + 8 weeks, $n = 10–14$; Tmx + 16 weeks, $n = 7–8$ of each animal group. *$P < 0.05$, **$P < 0.01$ and ***$P < 0.001$ by two-way ANOVA. Data are expressed as mean ± s.e.m.

known histological correlate of HHL[10,11] and synapse loss has been reported to occur in some demyelinating disorders, for example, in Charcot–Marie–Tooth (CMT) disease[25] and multiple sclerosis[26]. Therefore, we analysed the density of IHC ribbon synapses in control and demyelinated mice by immunostaining for the presynaptic ribbon and postsynaptic receptor patches using antibodies against CtBP2 and GluA2 (ref. 15). At 4 weeks post induction, when remyelination was complete but HHL was robust, IHC synapse density was the same in *DTA(+/−): Plp1/Cre(+/−)* and control cochleas at all regions examined (Fig. 6). This data indicates that transient loss of glial cells results in prolonged auditory deficits and HHL by a mechanism that is distinct from cochlear synaptopathy.

**HHL is associated with permanent heminodal disruption.** The observation that ABR P1 latency is longer after transient demyelination suggested that other aspects of myelin–axon interactions could be disrupted. Therefore, we examined whether the transient demyelination alters node of Ranvier density in auditory axons. Immunostaining for the paranodal protein Caspr[27] in combination with the myelin protein Plp1 showed that in the control P30 cochlea, nodes of Ranvier can be clearly observed along the ANFs in OSL (Fig. 7a), whereas heminodes formed by the first Schwann cells close to the IHCs can be seen at the habenula perforata (HP), the perforations in

the OSL through which auditory fibres pass on their way to and from the organ of Corti (Fig. 7a, see also Fig. 1a). Interestingly, the first 50 µm of the auditory nerve after the heminodes has a low density of nodes of Ranvier, indicating that consistent with previous reports[28,29], the myelin segments generated by the Schwann cells closest to the IHC are about that length. As expected for a severely demyelinated nerve, the appearance of nodes and heminodes was disrupted in *DTA(+/−): Plp1/Cre(+/−)* cochlea 1 week after DTA induction (Fig. 7b). Quantification of nodal density using co-immunostaining for Caspr and the nodal protein gliomedin[30] showed that 1 week after Schwann cell ablation, many nodes of Ranvier are lost in the OSL and the remaining ones have irregular appearance (Fig. 7c,d,i). Yet, consistent with the extensive remyelination, node of Ranvier density in the OSL completely regenerated by 4 weeks post induction, their appearance being indistinguishable from the controls (Fig. 7c–i).

In contrast, at the heminode zone and the area occupied by the first Schwann cells, Caspr and gliomedin localization did not recover normally. As shown in Fig. 7a,b, heminodes are dramatically disrupted 1 week after Schwann cell ablation compared with the control (Fig. 8a,b). In the control normal cochlea (Fig. 8a,c,e), the heminodal immunostaining is tightly clustered within the last 20 µm segments of the myelinated axon terminals (herein defined as the heminode zone). Although there is significant recovery of Caspr and gliomedin localization at

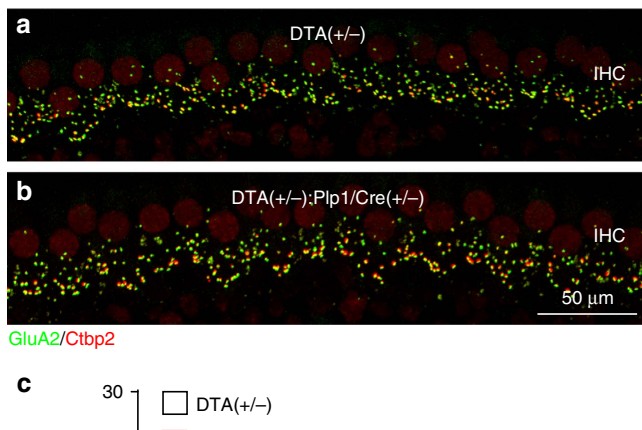

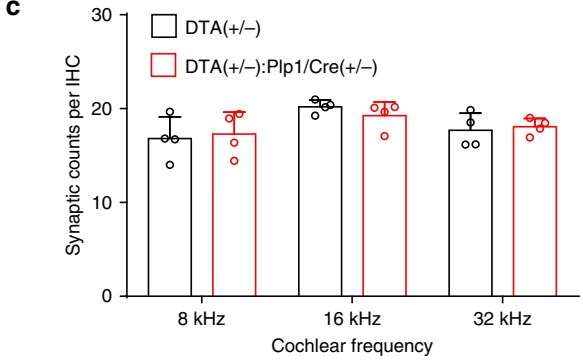

**Figure 6 | Transient cochlear Schwann cell ablation does not affect ribbon synapse density.** (**a,b**) Representative images of ribbon synapses immunostained with GluA2 (green) and Ctbp2 (red) from control (**a**) and $DTA(+/-):Plp1/Cre(+/-)$ (**b**) mice 4 weeks after tamoxifen induction. (**c**) Quantitative analyses of ribbon synapse counts per IHC at 8, 16 and 32 kHz cochlear regions. $n = 4$ of each animal group. Data are expressed as mean ± s.d.

heminodes during the postinduction period, it does not return to control levels 4 months after the insult (Fig. 8b,d,f). Moreover, aberrant localization of nodal proteins is observed proximal to the heminode zone (Fig. 8d,f, below dashed lines and Fig. 8g), whereas virtually no nodal proteins are present in that area in controls. Immunostaining for other nodal markers (sodium channels (NaV), ankyrin G (AnkG) and β4-spectrin) also shows disruption of heminodal organization in $DTA(+/-):Plp1/Cre(+/-)$ cochlea, further supporting the notion that Schwann cell–axon interactions in the area close to the HP are permanently affected by transient Schwann cell ablation (Supplementary Fig. 6). In contrast to the robust remyelination observed along the OSL, dysmyelinated fibres were observed at the axon segments closer to the HP. Immunostaining for neurofilament heavy chain (NFH), which labels unmyelinated axons[31], labelled nerve fibres below the HP in the $DTA(+/-):Plp1/Cre(+/-)$ cochlea but not in controls (Fig. 8h,i). Electron micrographs at the HP also illustrate the presence of unmyelinated fibres in $DTA(+/-):Plp1/Cre(+/-)$ mice but not in controls (Fig. 8j,k).

**HHL persists 1 year after Schwann cell ablation.** To determine whether the pathology observed 16 weeks after insult resolves with time, we studied a cohort of mice one year after DTA induction and found that the pathology remained unchanged. ABR thresholds remained normal in the $DTA(+/-):Plp1/Cre(+/-)$ mice (Supplementary Fig. 7a), indicating that the transient Schwann cell degeneration did not jeopardize long-term cochlea health. However, similar to that in the earlier time points, ABR P1 amplitudes, latencies and widths remained abnormal and indicative of auditory neuropathy (Supplementary Fig. 7b–d). Furthermore, node of Ranvier markers remained disorganized in the HP region (Supplementary Fig. 7g,h) but not at the OSL

(Supplementary Fig. 7e,f) in the $DTA(+/-):Plp1/Cre(+/-)$ cochlea. In addition, unmyelinated axon segments at the HP region were also present (Supplementary Fig. 7i,j). These results demonstrate compromised Schwann cell–axon interactions in this specific domain within the auditory nerve even 1 year after transient Schwann cell loss.

**Noise exposure causes HHL without affecting heminodes.** Exposure of rodents to noise levels higher than those that induce TTSs and HHL (110 versus 98 dB SPL) results in permanent hearing loss (permanent threshold shifts) and central auditory demyelination[32], raising the possibility that exposure to noise that induce TTS and HHL could also affect auditory nerve nodal structures. Therefore, we generated HHL in wild-type mice by exposing them to an octave-band noise (8–16 kHz, 98 dB SPL for 2 h)[4,33]. Two weeks after noise exposure, both ABR and DPOAE thresholds had recovered (Supplementary Fig. 8a,b), whereas the suprathreshold ABR P1 amplitudes were reduced at higher frequencies (Supplementary Fig. 8c). Consistent with previous reports[15,34], ABR P1 latencies were not significantly affected by noise exposure (Supplementary Fig. 8d), whereas ribbon synapse loss was evident at 32 kHz but not 11.3 kHz, consistent with a high-frequency HHL due to synaptopathy (Supplementary Fig. 8e–i). In contrast, the heminodes at the HP appeared normal (Supplementary Fig. 8j–m), suggesting that exposure to synaptopathic noise does not alter nodal organization.

**Noise- and demyelination-induced HHL are additive.** The results described above indicate that two different types of HHL exist: one caused by noise and involving synaptopathy, and the other caused by transient Schwann cell loss and involving nodal disruption. To test whether these two forms of HHL interact, we subjected mice that had undergone demyelination ($DTA(+/-):Plp1/Cre(+/-)$) at 3 weeks of age and their control counterparts ($DTA(+/-)$ mice) to noise exposure that causes synaptopathy at 8 weeks of age, and analysed their auditory function 2 weeks later (Fig. 9a). As shown above, 4 weeks after DTA-mediated Schwann cell ablation mice displayed HHL involving alterations in ABR Peak 1 amplitude and latency (Fig. 9b–d), which persisted even after noise exposure (Fig. 9e–g). Remarkably, noise exposure induced further decrease in ABR Peak 1 amplitudes beyond the one produced by demyelination (Fig. 9h–j) and the magnitudes of the shift in amplitudes were the same as in mice that had no demyelination (Fig. 9k–p). Furthermore, ABR P1 latencies remained unaffected after noise exposure in both demyelinated and control mice, indicating that the noise produced the same type of pathology independent of the prior demyelination (Fig. 9j,m,p). These findings indicate that exposure to synaptopathic noise causes additional degradation of neural output from the cochlea even when mice already suffer from HHL caused by transient Schwann cell loss. Thus, HHL caused by transient Schwann cell ablation is mechanistically distinct from that resulting from cochlear synaptopathy.

**Discussion**
Myelination of the peripheral nervous system (PNS) is important for normal conduction of APs and synchronized transmission of neural impulses[35,36], and damage to peripheral axons and their myelin sheaths causes many forms of peripheral neuropathy, such as CMT disease and Guillain–Barré syndrome (GBS)[37–39]. In this study, we uncover a new consequence of myelin pathology; that is, that transient Schwann cell loss results in permanent disruption of the cochlear heminodal structure and organization. We also discover that such disruption causes auditory neuropathy and HHL.

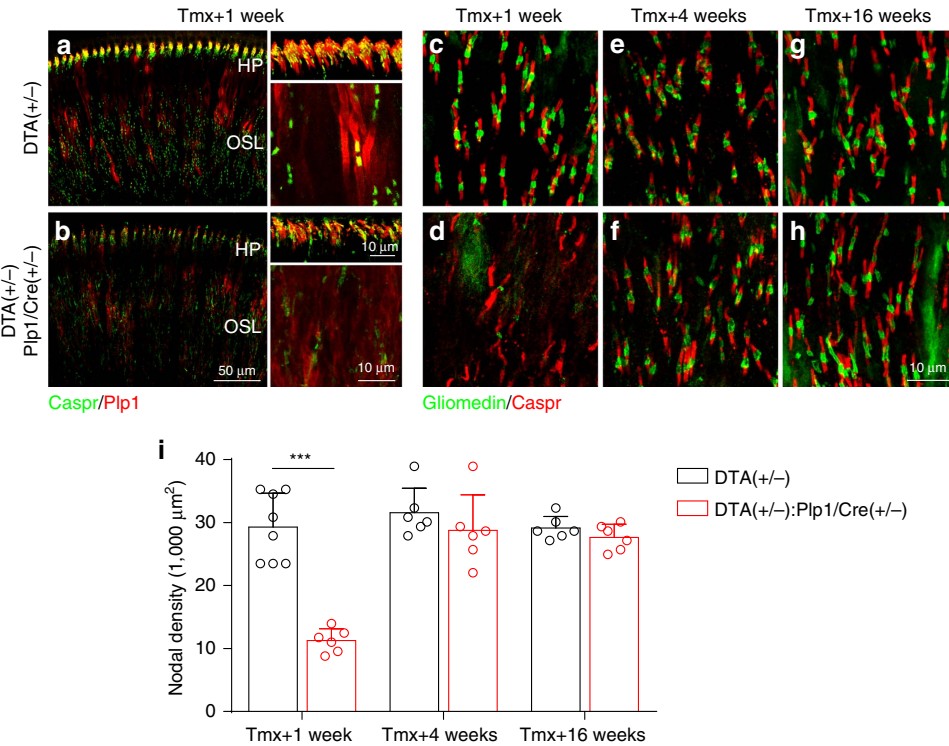

**Figure 7 | Transient Schwann cell ablation causes acute disruption of nodal structures followed by complete recovery of node of Ranvier density in ANFs in the OSL.** (**a,b**) Both cochlear heminodes and nodes of Ranvier are disrupted in $DTA(+/-):Plp1/Cre(+/-)$ mice at 1 week after tamoxifen induction. Tissues are immunostained for Plp1 (red) and Caspr (green). (**c–h**) Representative images showing that nodes of Ranvier at OSL are disrupted in $DTA(+/-):Plp1/Cre(+/-)$ cochlea at different times after tamoxifen treatment (**d,f,h**) compared with their corresponding controls (**c,e,g**). Nodes of Ranvier are lost in $DTA(+/-)$ mice 1 week after tamoxifen induction (**d**) but recover 4 weeks (**f**) and 16 weeks (**h**). (**i**) Node of Ranvier density quantification in $DTA(+/-):Plp1/Cre(+/-)$ and control mice at different times after tamoxifen treatment; $n = 6-8$ of each animal group. ***$P < 0.001$ by two-way ANOVA followed by Bonferroni's post test. Data are expressed as mean ± s.d.

Both cochlear type I afferents (SGNs) and medial olivocochlear efferents are myelinated[32]; however, it is unlikely that the auditory deficits we found after transient Schwann cell ablation are caused by damage to cochlear efferent nerves. First, sound-evoked medial olivocochlear activity is greatly depressed by anaesthesia during physiological recordings, thus should contribute little to our measurements of ABR amplitudes. Most importantly, complete lesion of cochlear efferent nerves fails to impair ABR peak 1 latencies (Maison *et al.*[40] and Liberman M.C., personal communications), further arguing against possible contribution of efferent nerves towards the auditory deficits observed in this study.

The bipolar SGNs form point-to-point connections between the sensory hair cells and neurons in the cochlear nucleus. At the SGN peripheral processes, the unmyelinated afferent terminals are tightly coupled to the ribbon synapses, whereas the myelinated axons extend from the HP to the SGN somata (Fig. 1a). Once the central SGN process enters the central nervous system (CNS), it becomes myelinated by oligodendrocytes. We used a mouse model of demyelination that results in loss of both Schwann cells in the peripheral nervous system and oligodendrocytes in the CNS. However, our physiological techniques allow us to study cochlear function without confounding issues from CNS demyelination by specifically analysing ABR peak 1. It has been shown both experimentally and computationally that ABR peak 1 is solely contributed by the IHCs and SGNs[41,42], which are myelinated by Schwann cells. In addition, the first heminode at the HP formed by Schwann cells is believed to be the initial spike generator of the SGNs[28,29,43]. Moreover, it has been shown that in this mouse model CNS neurons remyelinate robustly and recover function after DTA-mediated oligodendrocyte ablation[19], further excluding the involvement of central auditory components in this study. Thus, the physiological changes we observed after Schwann cell ablation are most likely due to alterations in the peripheral auditory components, that is, SGN synaptic terminal, axon or somata myelination or neuronal survival (refer to Fig. 1a).

Synaptic dysfunction is known to result in defects in SGN neural amplitudes and latencies[6,44]. However, synaptic density was not affected by Schwann cell ablation and demyelination. Although we did not examine whether synapses were transiently disrupted, it is unlikely to be that transient synaptopathy, if it occurred, could contribute to the persistent pathophysiology, as we have shown that when synapses regenerate after acute loss the auditory deficits recover as well[15]. As with the nerve terminals, the SGN cell bodies were also unaffected after transient loss of Schwann and satellite cells. Satellite cells express glutamine synthetase for converting ammonia and glutamate to glutamine[45], which is important for metabolizing excess glutamate and preventing excitotoxicity[46]. Satellite cells are also important for neuronal survival as knockout of sapB from these cells resulted in progressive neuronal death[47]. Complete regeneration of the spiral ganglion satellite cells in our study is consistent with the remarkable spontaneous proliferation and regeneration of satellite cells in other sensory ganglia[48–50]. Timely regeneration of the satellite cells and wrapping on SGN cell bodies may be important to preserve the survival and function of the SGNs.

In animal models of central and peripheral demyelinating diseases, many axons remyelinate but with significantly thinner myelin sheaths and shorter internodal lengths[51,52]. Intriguingly,

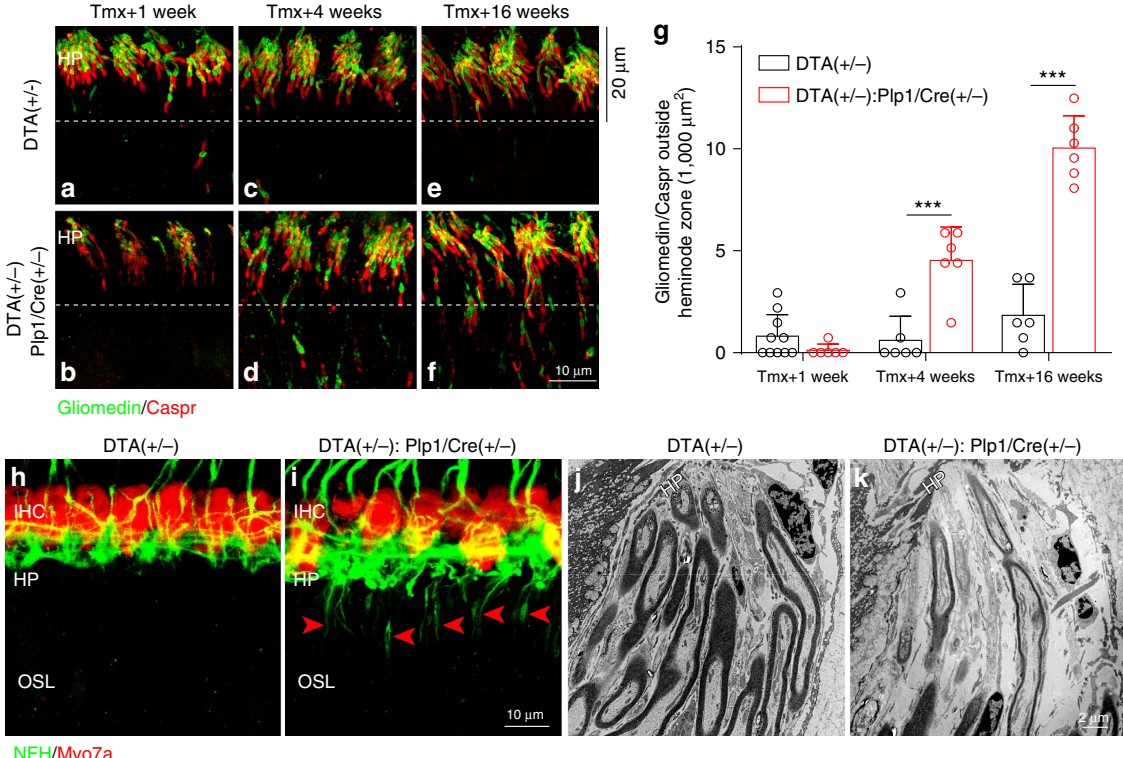

**Figure 8 | Transient Schwann cell ablation causes permanent disruption of the heminodes and dysmyelination at the HP.** (**a–f**) Heminodes at the habenula perforata (HP) remain disorganized in *DTA(+/−):Plp1/Cre(+/−)* cochlea (**b,d,f**) compared control cochlea (**a,c,e**) at 1 week (**a,c**), 4 weeks (**b,d**) and 16 weeks (**e,f**) after tamoxifen induction. (**g**) Increased number of aberrant nodal staining (co-localized Caspr/Gliomedin) is observed in *DTA(+/−):Plp1/Cre(+/−)* cochlea outside the pre-defined 20 μm heminode zone as shown in **a–f** (below dashed lines, $n = 6$–10 of each animal group). ***$P < 0.001$ by two-way ANOVA followed by Bonferroni's post test. Data are expressed as mean ± s.d. (**h,i**) Immunostaining for NFH, which preferentially stains unmyelinated axons, indicates the presence of dysmyelinated ANFs as they reach the HP (**i**, arrow heads) 16 weeks after tamoxifen induction in *DTA(+/−):Plp1/Cre(+/−)* cochlea. This staining is absent in control mice (**h**). HP, habenula perforata; IHC, inner hair cells; OSL, osseous spiral lamina. IHCs are stained for Myo7a. (**j,k**) Representative electron micrographs show dysmyelinated nerve fibres beneath the HP in *DTA(+/−):Plp1/Cre(+/−)* cochlea (**k**) but not in control cochlea (**j**).

after transient Schwann cell loss and massive axonal demyelination in response to DTA expression, we observed complete remyelination of ANFs at the OSL featuring normal myelin thickness and nodal density. It is worth noting that DTA induction results in damage to the myelinating glial cells without directly injuring their associated neurons. In contrast, in other animal models of demyelinating disorders axons may also be injured, for example, by nerve crush or inflammation[53,54]. These findings suggest that intact axons, but not injured ones, are capable of providing both physical and chemical cues sufficient for normal remyelination[55]. Our findings are consistent with a previous report in which complete remyelination was observed in sciatic nerve after selective ablation of myelinating glial cells[56]. A recent report shows that in the CNS, remyelination and normal nodal length can be completely restored on small caliber axons, suggesting that complete myelin recovery may be dependent on axon caliber[57]. Thus, an alternative scenario is that complete remyelination of the ANFs can be attributed to their smaller axon caliber (∼1.5 μm diameter) compared with other peripheral nervous system axons, such as ventral roots and quadriceps nerves (>3 μm diameter)[58,59].

Analysis of node of Ranvier markers indicated specific and persistent disruption of the heminodes at the HP, but with complete recovery of nodes along the axonal tracts. Emerging evidence indicates that AP generation in the auditory nerve occurs at the heminode[28,29,43] and the clustering of mature heminodes may be important for tight coupling of IHC synaptic

transmission and ANF spike generation[29]. The peak 1 of the ABR waveform, which reflects the gross ANF discharge, relies on concerted nerve conduction from the unmyelinated nerve terminals (inner spiral plexus), the heminodes (the spike generator) and the nodes of Ranvier[29]. The inner spiral plexus is unlikely to be involved in the observed pathophysiology, because the precisely timed short-latency nerve conduction is ensured by multiple mechanisms[43]. The complete recovery of nodes of Ranvier also precludes their contributions. As the highly clustered heminodes allow generation of synchronized AP spikes in multiple unisynaptic ANFs[28,29,43], we propose that AP dyssynchronization resulting from heminodal disruption is the main contributor to the loss of temporal resolution in the waveform and therefore to the deterioration of peak 1 amplitude, latency and width.

The postsynaptic heminodes are typically clustered about 20 μm away from the IHC synaptic zone and are enriched in ion channels[28,29]. Recent studies indicate that these structural and molecular features of auditory nerve nodes and heminodes mature after the onset of hearing in rats[29]. Specifically, during the early postnatal ages (<P17), the immature heminodes lack Kv1.1 channels and are irregularly positioned along individual axons with about 60 μm spread from the HP[29]. Interestingly, the pathological heminodes observed in this study closely resembled those seen before hearing onset[29]. Therefore, it appears that during the process of Schwann cell regeneration, remyelination of ANFs follows a path similar to the one taking place during

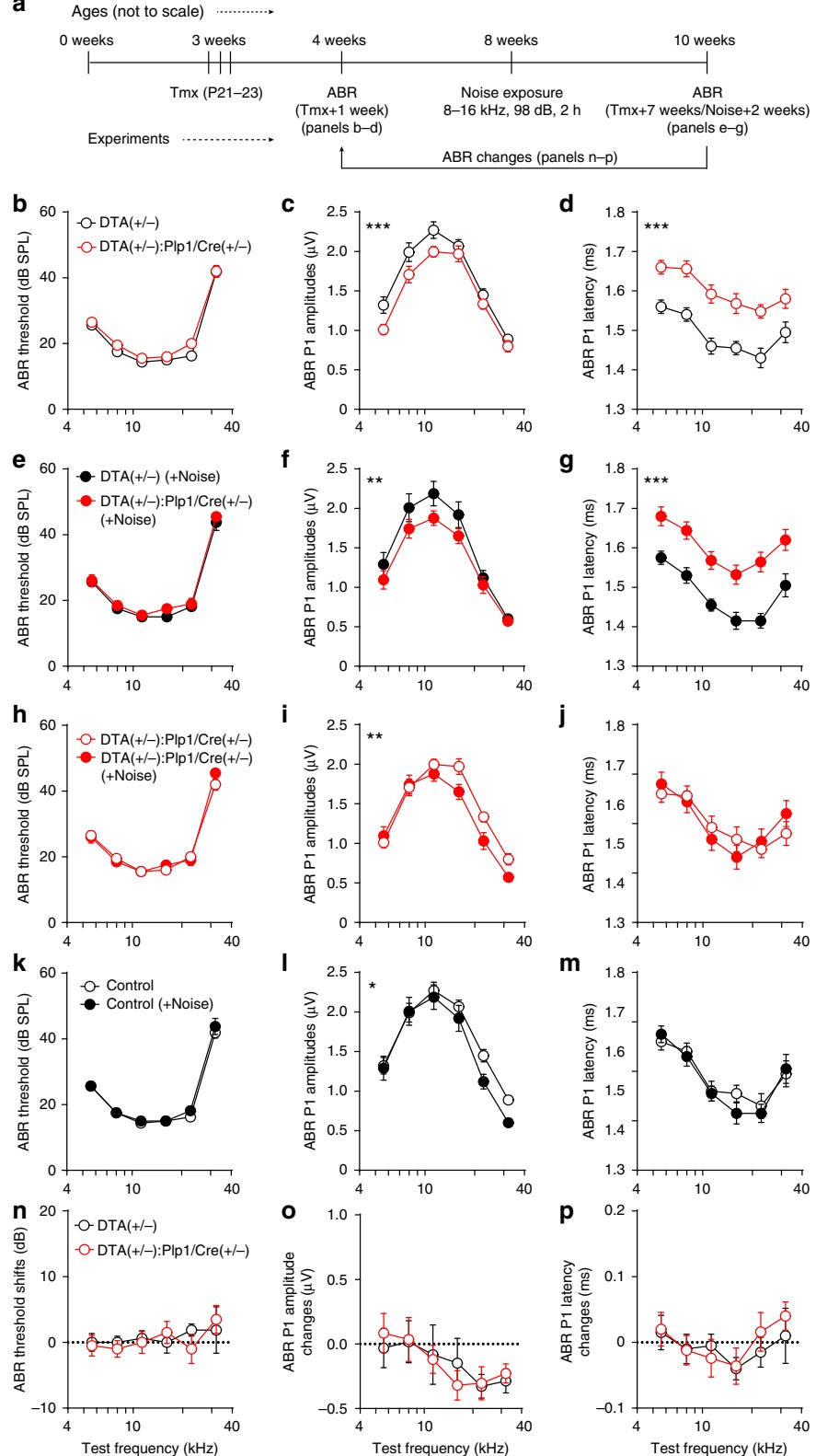

**Figure 9 | Synaptopathic noise exposure causes further hidden hearing loss in mice that had transient Schwann cell ablation.** (**a**) Flow chart showing the experimental design. (**b–d**) Tamoxifen treatment induces hidden hearing loss characterized by normal ABR thresholds (**b**), lower ABR P1 amplitudes (**c**) and longer P1 latencies (**d**) in *DTA( + / − ):Plp1/Cre( + / − )* mice. (**e–g**) Hidden hearing loss caused by transient Schwann cell loss persists after noise exposure (8–16 kHz, 98 dB SPL for 2 h). Noise exposure induces further hidden hearing loss in mice with established pathology after Schwann cells loss (**h–j**) and the control mice (**k–m**). (**n–p**) The magnitude of the noise-induced change in *DTA( + / − ):Plp1/Cre( + / − )* mice is the same as that on *DTA( + / − )* mice. For **n–p**, physiological data after noise exposure were normalized to that before noise exposure within each genotype. ABR P1 amplitudes and latencies were analysed at 70 dB SPL. $n = 8$–10 of each group. $*P < 0.05$, $**P < 0.01$ and $***P < 0.001$ by two-way ANOVA. Data are expressed as mean ± s.e.m.

early postnatal development, but remyelination and heminode maturation at the HP fail to proceed, possibly due to either the presence of inhibitory signals in the mature cochlea or lack of promoting factors that are present during development. As heminodes and nodes share similar molecular microdomain structure[28,29] and nodes of Ranvier recovere completely, the observed nodal pathology is likely to be due to the physical–chemical micro-environments at the HP, rather than a generic axoglial signal guiding the nodal assembly.

HHL has emerged as a major health concern[13]. HHL is defined and diagnosed by abnormal ABR amplitudes and latencies but normal thresholds[9–11], thus undetectable by standard audiometry. Subjects with this type of hearing loss may have difficulties in speech discrimination and temporal processing, particularly in a noisy environment[13]. Until now, synaptic loss or cochlear synaptopathy resulting from either noise exposure or ageing was the only known cause of HHL[5,6] and exploration of therapeutic approaches for HHL has been focused on repairing synapses[15,33]. Our findings indicate that HHL can also arise from pathologies of cochlear Schwann cell (schwannopathy) or nodal structures (nodopathy), rather than synaptopathy. Indeed, some patients with demyelinating disorders such as CMT disease and GBS have auditory deficits, associated with reduced ABR amplitudes and increased latencies[60–64]. Our animal model shares mechanistic similarities with GBS, where Schwann cells are specifically and transiently damaged[52]. Our findings that Schwann cell ablation does not affect ABR or DPOAE thresholds are consistent with rare reports of hearing threshold elevations in GBS patients[63,64]. Intriguingly, in those patients who suffered acute hearing loss, although hearing threshold gradually recovered, their ABR waveforms showed persistent increase in latency[62]. Therefore, an even greater proportion of GBS patients may have HHL that has not been diagnosed clinically. Given the growing prevalence of Zika infections, which has been shown to correlate with an increase in GBS patients worldwide[65,66], the prevalence of HHL related to acute peripheral demyelinating disorders is likely to increase. Thus, the cochlear neuropathy revealed in our animal model may have clinical implications in diagnosis and eventual treatment of the different types of HHL.

## Methods

**Animals.** *Plp1/CreER^T* (*Plp1/Cre*; stock number 005975), *Ai14:Rosa26^tdTomato* (*tdTomato*; stock number 007908) and *Rosa26^DTA* (*DTA*; stock number 006331) transgenic mice of both genders were obtained from The Jackson Laboratory. *Plp1/eGFP* mice were obtained from Wendy Macklin at the University of Colorado, Denver. To express DTA in cochlear myelinating glial cells, we crossed *Plp1/Cre(+/−)* with *DTA(+/+)* mice. *Cre*-negative littermates were used as controls. For lineage-tracing experiments, *Plp1/Cre(+/−):tdTomato(+/−)* or *Plp1/Cre(+/−):Plp1/eGFP(+/−)* mice were crossed with *DTA(+/−)* mice. Offsprings carrying all three alleles were used for experiments and *DTA*-negative littermates were used as controls. All mice used in this work were on a mixed background containing C57Bl6 and FVB/N strains, and mice tested for auditory physiology were heterozygous for ahl allele. Both males and females were used in this study. The genotypes of the control and experimental animals were blinded to the investigators until the completion of data acquisition and analysis. All animal procedures were approved by the Institutional Animal Care and Use Committee of University of Michigan.

**Tamoxifen and EdU injections.** Tamoxifen (Sigma, T5648) was dissolved in corn oil (Sigma, C8267) at 37 °C to obtain 10 mg ml$^{-1}$ working solution. Mice were injected i.p. with tamoxifen at 50 mg kg$^{-1}$ per day from P21 to P23. Animals were analysed at 1 week to 1 year after the last tamoxifen injection. To examine cell proliferation, P29 mice were i.p. injected with EdU (obtained from the Click-iT EdU Alexa Fluor 488 Imaging Kit (Thermo Fisher Scientific, C10337)) once at 10 mg kg$^{-1}$ dosage. Mice were killed and tissue analysed at P30. EdU staining was performed following the manufacturer's instructions.

**ABR and DPOAE analyses.** ABRs and DPOAEs were performed on mice anaesthetized with xylazine (20 mg kg$^{-1}$, i.p.) and ketamine (100 mg kg$^{-1}$, i.p.). For ABR recordings, needle electrodes were placed into the skin (a) at the dorsal midline close to the neural crest, (b) behind the left pinna and (c) at the base of the tail (for a ground electrode). ABR potentials were evoked with 5 ms tone pips (0.5 ms rise–fall, with a cos2 envelope, at 40 s$^{-1}$) delivered to the eardrum at log-spaced frequencies from 5.6 to 32 kHz. The response was amplified (10,000 ×) and filtered (0.3–3 kHz) with an analog-to-digital board in a PC-based data-acquisition system. Sound level was raised in 5 dB steps from 10 to 80 dB SPL. At each level, 1,024 responses were averaged (with stimulus polarity alternated) after 'artefact rejection.' The DPOAEs in response to two primary tones of frequency f1 and f2 were recorded at $(2 \times f1) - f2$, with f2/f1 = 1.2, and the f2 level 10 dB lower than the f1 level. Ear-canal sound pressure was amplified and digitally sampled at 4 μs intervals. DPOAE thresholds were defined as the f1 level required to produce a response at −10 dB SPL. Both ABR and DPOAE recordings were performed using EPL cochlear function test suite (Mass Eye and Ear, Boston, MA, USA). ABR peak amplitudes, latencies and widths were analysed with excel and ABR peak analysis software (Mass Eye and Ear).

**Noise exposure.** Awake wild-type FVB/N mice or *DTA(+/−):Plp1/Cre(+/−)* and their control mice were placed within small cells in a subdivided cage, suspended in a reverberant noise exposure chamber. FVB/N mice (16 weeks) were exposed to an octave band of noise (8–16 kHz) at 100 dB SPL for 2 h. *DTA(+/−):Plp1/Cre(+/−)* and control mice (8 weeks) were exposed to the same noise at 98 dB SPL for 2 h. Calibration of noise was performed immediately before each noise overexposure to ensure that SPL levels varied by <1 dB across the cages.

**Immunohistochemistry and confocal microscopy.** Inner ear tissues were dissected and fixed in 4% paraformaldehyde in 0.1 M phosphate buffer for 30 min–2 h at room temperature, followed by decalcification in 5% EDTA at 4 °C for 5 days. Cochlear tissues were microdissected and permeabilized by freeze–thawing in 30% sucrose. The microdissected tissues were incubated in blocking buffer containing 5% normal horse serum and 0.3% Triton X-100 in PBS for 1 h. Tissues were then incubated in primary antibodies (diluted in blocking buffer) at 37 °C for 16 h. The primary antibodies used in this study were as follows: anti-myosin VIIa[15] (rabbit anti-Myo7a; Proteus Biosciences, Ramona, CA; 1:500), anti-Sox2 (goat anti-Sox2; sc-17320, Santa Cruz Biotechnologies, Santa Cruz, CA; 1:200)[16], anti-Sox10 (goat anti-Sox10; sc-17342, Santa Cruz Biotechnologies; 1:200)[67], anti-MBP (rat anti-MBP; MAB386, EMD Millipore, Billerica, MA; 1:1,000)[67], anti-carboxy-terminal binding protein 2 (mouse anti-Ctbp2 IgG1; BD Biosciences, San Jose, CA; 1:200)[15], anti-glutamate receptor 2 (mouse anti-GluA2 IgG2a; Millipore, Billerica, MA; 1:2,000)[15], anti-NFH (chicken anti-NFH; AB5539, EMD Millipore; 1:2,000)[31], anti-Plp1 (gift of Wendy Macklin; 1:500)[16], anti-Caspr (gift of Roman Giger; 1:1,000)[68], anti-gliomedin (gift of Elior Peles; 1:500)[30], anti-pan sodium channel (anti-NaV; gift of Roman Giger; 1:75)[68], anti-β4-spectrin and anti-Ankyrin G (gifts of Paul Jenkins; 1:500)[69]. Tissues were then incubated with appropriate Alexa Fluor-conjugated fluorescent secondary antibodies (Invitrogen, Carlsbad, CA; 1:500 in blocking buffer) for 1 h at room temperature. Nuclei were labelled with either 4,6-diamidino-2-phenylindole (1 μg ml$^{-1}$; Life Technologies) or Hoechst 33342 (1:2,000; Life Technologies). The tissues were mounted on microscope slides in ProLong Gold Antifade Mountant (P36930, Thermo Fisher Scientific). Unless otherwise stated, mid-apical cochlear tissues (8–16 kHz region) were used for analyses.

Confocal z-stacks (0.3 μm step size) of cochlear tissues were taken using either a Zeiss LSM710 or a Leica SP8 microscope equipped with ×40 and ×63 oil-immersion lens. Images for quantitative analyses of the nodal structures were taken under ×63 lens with ×5 optical zoom. Imaging and analyses of cochlear ribbon synapses were performed as previously described[15]. ImageJ software (version 1.46r, NIH, MD) was used for image processing and three-dimensional reconstruction of z-stacks. All immunofluorescence images shown are representative of at least three individual mice each group.

**Plastic sections and transmission electron microscopy.** Mice were perfused intracardially with 4% paraformaldehyde in 0.1 M phosphate buffer. Cochleae were then extracted and postfixed with 1.5% paraformaldehyde and 2.5% glutaraldehyde, followed by osmification in 1% osmium tetroxide, decalcification in 5% EDTA, dehydration in ethanol and embedding in araldite resin. The embedded cochleae were hardened at 60 °C for 5 days before semi-thin (1 μm) plastic sectioning for light microscopy and ultra-thin (70 nm) sectioning for transmission electron microscopy using a Leica UC6 ultramicrotome. The semi-thin sections were mounted on microscope slides in Permount (Fisher Scientific) and the SGN cell bodies imaged under light microscope using ×63 lens.

Ultra-thin sections were sequentially poststained with 6% w/v uranyl acetate and 4.4% w/v lead citrate. Transmission electron microscopy was performed on Jeol 1400-plus electron microscope (Jeol USA, Peabody, MA). Multiple non-overlapping regions of the ANF cross-sections were imaged at ×3,400 magnification. The circumference of each axon fibre and axon fibre plus myelin sheath was measured using ImageJ software. Axon diameters were estimated as circumference/π and the g-ratio was calculated as axon diameter / (axon + myelin sheath diameter). Axons with circularity ($4 \times \pi \times$ area/perimeter2) under 6 were

excluded from analysis. All images of semi-thin sections and electron microscope sections shown are representative of at least three individual mice each group.

**Statistical analysis.** Statistical tests were performed using Graphpad Prism 6 (Graphpad Software Inc., La Jolla, CA). Distribution of the data was assumed to be normal, but this was not formally tested. No statistical methods were used predetermine sample sizes, but our sample sizes are similar to those reported in our previous publications[15,16,70]. Statistical differences in auditory physiology, ribbon synapse counts, densities of myelinated axons, densities of nodes and aberrant nodes were analysed using two-way ANOVA, followed by Bonferroni's multiple comparisons test. Densities of SGN cell bodies and percentages of SGN cell bodies wrapped by satellite cells were compared using unpaired Student's *t*-test. $\chi^2$-test and frequency distribution plots were used to compare the distribution of axon diameters and g-ratios (axon diameter/(axon + myelin sheath diameter) (300–500 axons from 3 mice per group). Student's *t*-test was also used to compare the average axon diameter and g-ratio of each mouse.

**Data availability.** All relevant data are included in the manuscript or its Supplementary Information and available from the authors upon request.

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

### Acknowledgements

We thank Roman Giger (University of Michigan) for Caspr and NaV antibodies, Paul Jenkins (University of Michigan) for AnkG and β4-spectrin antibodies, Elior Peles (Weizmann Institute of Science) for gliomedin antibodies, Angelica Gigliello for technical assistance and staff at the University of Michigan Microscopy & Image Analysis Laboratory core for help in electron microscopy. We thank M. Charles Liberman, Jochen Schacht, Susan Shore, James Teener and Roman Giger for critical reading of the manuscript. This study is supported by NIH/NIDCD R01DC004820 (to G.C.) and Hearing Health Foundation (to G.W.).

### Author contributions

G.W. and G.C. designed the study. G.W. performed the experiments and analysed the data. G.W. and G.C. interpreted the data and wrote the manuscript.

### Additional information

**Competing financial interests:** G.C. is a scientific founder of Decibel Therapeutics, has an equity interest in and has received compensation for consulting. The company was not involved in this study. G.W. declares no competing financial interest

