## [Peer Review File · Nature Communications]

REVIEWERS' COMMENTS:

Reviewer #1 (Remarks to the Author):

The authors' clarification on the current understanding of what constitutes HHL in humans is appreciated. Given the dearth of diagnostic criteria for HHL, the authors' response to this issue is satisfactory. However, it's still case that the data presented in this manuscript does not directly support the claim that demyelination could be responsible for HHL cases in human patients. This observation of abnormal ABR findings in the setting of acute Schwann cell ablation could perhaps be better described as a new animal model of HHL that suggests that demyelination is a potential mechanism for HHL, rather than stating that it is a novel mechanism for HHL observed in human patients.

Regarding the non-specificity of DTA-mediated ablation using the a Plp1-driven inducible Cre: While it's true that the ABR peak 1 is solely contributed to by spiral ganglion neurons, it's not inconceivable that central demyelination could affect the response of these cells or modulate auditory input, for example through efferent CN VIII activity onto outer hair cells. It would strengthen the manuscript if the authors would include a discussion with these kinds of possibilities. In general, it would also be informative to comment on the overall health of the mice used.

Reviewer #2 (Remarks to the Author):

The revised manuscript "Transient auditory nerve demyelination, a new mechanism for hidden hearing loss" Guoqiang Wan and Gabriel Corfas takes care of many of the points raised by the referees.

- The manuscript shows that experimentally induced de- and re-myelination is incomplete in auditory nerve. That is an interesting finding for audiologist, no question.
- The incomplete de-myelination has consequences for propagation of action potentials in the auditory nerve. This is nice to see but not at all surprising.
- The results may (probably do) explain the ABR-abnormalities reported in a few cases-studies of obvious re-myelination after diagnosis of Guillain-Barré syndrome. This is relevant for audiologists and neurologists working with patients diagnosed with de-myelination diseases.

For more specialized journals (whether in the glia-field or auditory field) I would not hesitate to recommend publication as it is (again, beautiful data!).

However, I have still trouble to see why this would be of any general interest deserving a broad readership.

Reviewer #3 (Remarks to the Author):

MAJOR COMMENTS

As I said when I reviewed this paper previously, it is an excellent piece of work that clearly makes a case for demyelination as a possible mechanism for HHL (as defined by electrophysiological data, well discussed in the authors' responses to previous Revs 1 and 2). The authors have done a nice job of making a case for the possible relevance of demyelination, Zika, GB, etc.

Very nice data on the growth of ABR amplitude and latency have been added. This show the surprising result that the amplitude effect is quite small, in that an increase in the signal level of only a few dB could restore amplitudes to normal (Fig. S4). By contrast the latency effect is very robust, at least 20 dB in the stimulus domain (Fig. S5). This result, in my mind, suggests that the major effect of demyelination is in the temporal, and not the conduction or conduction block domain, an idea supported by the increase in the width of the ABR peak. The authors might consider pointing this out in a more concerted way than currently and discussing it as a basis for their currently rather poorly supported discussion of dyssynchrony (see the next comment).

line 272 - "dyssynchrony", not clear what is meant by this. Can you point to a figure that provides a basis for this. Later, on line 331-332, it is stated that increased P1 latencies and decreased amplitudes are correlated with dyssynchrony. It's not clear why. What I'm asking for here is just a little uncertainty about dyssynchrony in your preparation, which hasn't been measured.

In Fig. 9 n-p, a normalization was applied to the data that is somewhat obscure and unjustified. It has the problem that the result shows no change in amplitude of the ABR, for example, at 10 kHz in the difference plot (o), whereas there clearly are (small) ones in the data plots (i and l). It seems to me that this data presentation would be more transparent and perhaps more meaningful if plotted as simple differences, i.e. $n = \Delta h - \Delta k$, $o = \Delta i - \Delta l$, etc.

Regardless of how the data are plotted, the results in Fig. 9 are based on small differences, for some reason smaller than expected from the previously shown physiological data, especially for the amplitude data. Is there a reason why the DTA effect was smaller for Fig. 9i than in Fig. 4A?

MINOR COMMENTS

line 90 - probably should be ". . . supporting cells in the organ of Corti."

line 91 - not clear what "from" means here. Maybe "with"?

line 559 - "mice"

Reviewer #1 (Remarks to the Author):

The authors' clarification on the current understanding of what constitutes HHL in humans is appreciated. Given the dearth of diagnostic criteria for HHL, the authors' response to this issue is satisfactory. However, it's still case that the data presented in this manuscript does not directly support the claim that demyelination could be responsible for HHL cases in human patients. This observation of abnormal ABR findings in the setting of acute Schwann cell ablation could perhaps be better described as a new animal model of HHL that suggests that demyelination is a potential mechanism for HHL, rather than stating that it is a novel mechanism for HHL observed in human patients.

RESPONSE: We appreciate the reviewer for general support of our revision and clarification. We agree with the reviewer that at this time our findings are based on mice. Therefore, we have relegated any discussion of potential human relevance to the last paragraph on the discussion, where we make it clear that this is a possibility that needs to be tested.

Regarding the non-specificity of DTA-mediated ablation using the a Plp1-driven inducible Cre: While it's true that the ABR peak 1 is solely contributed to by spiral ganglion neurons, it's not inconceivable that central demyelination could affect the response of these cells or modulate auditory input, for example through efferent CN VIII activity onto outer hair cells. It would strengthen the manuscript if the authors would include a discussion with these kinds of possibilities.

RESPONSE: We thank the reviewer for the excellent point. We now discuss the reasons why we believe it is unlikely that efferents contribute to the demyelination-induced HHL. As we state now in the last paragraph page 12:

“Both cochlea type I afferents (SGNs) and medial olivocochlear (MOC) efferents are myelinated³², however, it is unlikely that the auditory deficits we found after transient Schwann cell ablation are caused by damage to cochlear efferent nerves. First, sound-evoked MOC activity is greatly depressed by anesthesia during physiological recordings, thus should contribute little to our measurements of ABR amplitudes. Most importantly, complete lesion of cochlear efferent nerves fails to impair ABR peak 1 latencies (Maison S.F. et al.⁴⁰ and Liberman MC. personal communications), further arguing against possible contribution of efferent nerves towards the auditory deficits observed in this study.”

In general, it would also be informative to comment on the overall health of the mice used.

RESPONSE: we have also included a statement regarding the health of the mice (see paragraph 2 of the Results section).

“As anticipated from prior studies of mice with DTA-induced ablation of Plp1 expressing cells¹⁹, tamoxifen i.p. injection to mice carrying Plp1/CreERT transgene and Rosa26DTA alleles [DTA(+/-):Plp1/Cre(+/-)] from P21-23 resulted in uncoordinated gait and hind-limb paralysis between 1-2 weeks post DTA expression followed by complete recovery of normal gait by 3-4 weeks post DTA expression. This is consistent with the robust regenerative capacity of Plp1-expressing glia cells¹⁹. Interestingly, while DTA induction at P35 results in significant mortality 3 weeks later (¹⁹ and our unpublished data), we found that ablation of Plp1-expressing cells at P21 does not result in lethality. These observations suggest that the health effects of loss of Plp1+ cells increase with age.”

Reviewer #2 (Remarks to the Author):

The revised manuscript "Transient auditory nerve demyelination, a new mechanism for hidden hearing loss" Guoqiang Wan and Gabriel Corfas takes care of many of the points raised by the referees.

- The manuscript shows that experimentally induced de- and re-myelination is incomplete in auditory nerve. That is an interesting finding for audiologist, no question.
- The incomplete de-myelination has consequences for propagation of action potentials in the auditory nerve. This is nice to see but not at all surprising.
- The results may (probably do) explain the ABR-abnormalities reported in a few cases-studies of obvious re-myelination after diagnosis of Guillain-Barré syndrome. This is relevant for audiologists and neurologists working with patients diagnosed with de-myelination diseases.

For more specialized journals (whether in the glia-field or auditory field) I would not hesitate to recommend publication as it is (again, beautiful data!).

However, I have still trouble to see why this would be of any general interest deserving a broad readership.

RESPONSE: We are grateful to the reviewer for favorable assessment of our revision and quality of data. We believe this study warrants a broad readership because of the growing recognition in the prevalence of HHL and its previously unappreciated influence in hearing perception. Cochlear synaptopathy, which has been synonymous to HHL-causing neuropathy previously, is now only one of the causes of HHL. Moreover, with the differential diagnosis of HHL presented in this study, it is also a timely work that guides clinical assessment of HHL in human patients suffering demyelination or other hearing problems, and thus should be highly relevant to both neurologists, otolaryngologists and audiologists.

Reviewer #3 (Remarks to the Author):

MAJOR COMMENTS

As I said when I reviewed this paper previously, it is an excellent piece of work that clearly makes a case for demyelination as a possible mechanism for HHL (as defined by electrophysiological data, well discussed in the authors' responses to previous Revs 1 and 2). The authors have done a nice job of making a case for the possible relevance of demyelination, Zika, GB, etc.

Very nice data on the growth of ABR amplitude and latency have been added. This show the surprising result that the amplitude effect is quite small, in that an increase in the signal level of only a few dB could restore amplitudes to normal (Fig. S4). By contrast the latency effect is very robust, at least 20 dB in the stimulus domain (Fig. S5). This result, in my mind, suggests that the major effect of demyelination is in the temporal, and not the conduction or conduction block domain, an idea supported by the increase in the width of the ABR peak. The authors might consider pointing this out in a more concerted way than currently and discussing it as a basis for their currently rather poorly supported discussion of dyssynchrony (see the next comment).

line 272 - "dyssynchrony", not clear what is meant by this. Can you point to a figure that provides a basis for this. Later, on line 331-332, it is stated that increased P1 latencies and decreased amplitudes are correlated with dyssynchrony. It's not clear why. What I'm asking for here is just a little uncertainty about dyssynchrony in your preparation, which hasn't been measured.

RESPONSE: The reviewer raised two related and insightful comments on how demyelination at heminodes may alter ABR peak 1 amplitude and latency. First, we agree with the reviewer that temporal impairment rather than blockade of nerve conduction is the likely culprit for HHL, as we have discussed in the previous Response. Second, we agree that the word “dyssynchrony” can be confusing. We have changed the text to express our conclusions and ideas more clearly. We believe that highly clustered heminodes allow generation of synchronized action potentials in the multiple auditory nerve fibers (ANFs) innervating each inner hair cell. Because ABR peak 1 waveform is a function of the ANF discharge, dyssynchronization of the action potentials as a result of heminodal disruption should lead to loss of temporal resolution in the waveform and deterioration of both amplitude and latency. To clarify our interpretation, we have removed the original sentence questioned by the reviewer (line 329-332) and revised as below (page 15):

“The peak 1 of the ABR waveform, which reflects the gross ANF discharge, relies on concerted nerve conduction from the unmyelinated nerve terminals (inner spiral plexus), the heminodes (the spike generator) and the nodes of Ranvier²⁹. The inner spiral plexus is unlikely involved in the observed pathophysiology because the precisely timed short-latency nerve conduction is ensured by multiple mechanisms⁴³. The complete recovery of nodes of Ranvier also precludes their contributions. As the highly clustered heminodes allow generation of synchronized AP spikes in multiple unisynaptic ANFs^{28, 29, 43}, we propose that AP dyssynchronization resulting from heminodal disruption is the main contributor to the loss of temporal resolution in the waveform and therefore to the deterioration of both peak 1 amplitude, latency and width.”

In Fig. 9 n-p, a normalization was applied to the data that is somewhat obscure and unjustified. It has the problem that the result shows no change in amplitude of the ABR, for example, at 10 kHz in the difference plot (o), whereas there clearly are (small) ones in the data plots (i and l). It seems to me that this data presentation would be more transparent and perhaps more meaningful if plotted as simple differences, i.e. $n = \Delta h - \Delta k$, $o = \Delta i - \Delta l$, etc.

RESPONSE: We have revised the figure, specifically panel (o), as per reviewer’s recommendation. The changes in P1 amplitude are now in absolute values (μV) instead of percentage. The conclusion remains the same.

Regardless of how the data are plotted, the results in Fig. 9 are based on small differences, for some reason smaller than expected from the previously shown physiological data, especially for the amplitude data. Is there a reason why the DTA effect was smaller for Fig. 9i than in Fig. 4A?

RESPONSE: Different batches of mice were used for Fig.9 and Fig.4, nevertheless we could compare the ABR results at *Tmx + 1wk* time point of the two batches: Fig.4a vs Fig.9c. For a thorough comparison, we plotted all 4 groups together (Figure below). As the reviewer can see, there is no difference between Fig.9c and Fig.4a in either control groups or demyelination groups (two-way ANOVA). Most importantly, within each batch or each experiment, there is reproducible and significant difference between control and Plp-Cre:DTA mice (two-way ANOVA). Small variations in the absolute changes of amplitudes may arise from differences in calibration of acoustic systems and other factors; however, within each cohort of animals (we use littermates for control and experimental groups), demyelination-induced HHL is highly reproducible.

MINOR COMMENTS

line 90 - probably should be “. . supporting cells in the organ of Corti.”

line 91 - not clear what “from” means here. Maybe “with”?

line 559 - “mice”

RESPONSE: We have corrected the errors.